# Hypothyroidism following Radiotherapy for Head and Neck Cancer: A Systematic Review of the Literature and Opportunities to Improve the Therapeutic Ratio

**DOI:** 10.3390/cancers15174321

**Published:** 2023-08-29

**Authors:** Michael K. Rooney, Lauren M. Andring, Kelsey L. Corrigan, Vincent Bernard, Tyler D. Williamson, Clifton D. Fuller, Adam S. Garden, Brandon Gunn, Anna Lee, Amy C. Moreno, William H. Morrison, Jack Phan, David I. Rosenthal, Michael Spiotto, Steven J. Frank

**Affiliations:** Department of Radiation Oncology, MD Anderson Cancer Center, Houston, TX 77030, USAvbernard@mdanderson.org (V.B.); twilliamson@mdanderson.org (T.D.W.); sjfrank@mdanderson.org (S.J.F.)

**Keywords:** hypothyroidism, head and neck cancer, radiotherapy

## Abstract

**Simple Summary:**

Hypothyroidism is a common toxicity experienced after radiotherapy for head and neck malignancies. In this systemic review of the literature, we consolidated the body of existing evidence regarding the incidence and risk factors for radiotherapy-induced hypothyroidism. We found large variations in estimates, although the most recent large investigations suggest an overall incidence of approximately 40–50%, which is significantly higher than many historical estimates. Furthermore, we synthesized the evidence regarding the dosimetric-based prediction of hypothyroidism, including a summary of various proposed dosimetric parameters that may be utilized clinically to optimize plans. We found a paucity of literature investigating the use of modern advanced radiotherapy techniques, such as particle therapy or MRI-guided radiotherapy, to reduce the risk of thyroid toxicity. These findings may be useful to guide clinicians and researchers aiming to improve the care of patients undergoing radiotherapy for head and neck cancer.

**Abstract:**

(1) Background: Radiotherapy (RT) is a central component for the treatment of many head and neck cancers. In this systematic review of the literature, we aimed to characterize and quantify the published evidence on RT-related hypothyroidism, including estimated incidence, clinical risk factors, and dosimetric parameters that may be used to guide clinical decision making. Furthermore, we aimed to identify potential areas of improvement in the prevention and clinical management of RT-induced hypothyroidism, including the role of modern advanced therapeutic techniques. (2) Methods: We conducted a systemic review of the literature in accordance with Preferred Reporting Items for Systematic Reviews and Meta-Analysis (PRISMA) guidelines. PubMed and Google Scholar were searched to identify original research articles describing the incidence, mechanism, dosimetry, treatment, or prevention of radiation-related hypothyroidism for adults receiving RT for the treatment of head and neck cancers. The snowball method was used to identify additional articles. For identified articles, we tabulated several datapoints, including publication date, patient sample size, estimated hypothyroidism incidence, cancer site/type, follow-up period, radiation modality and technique, use of multimodality therapy, method of thyroid function evaluation, and proposed dosimetric predictors of hypothyroidism. (3) Results: One hundred and eleven articles met inclusion criteria, reflecting a range of head and neck cancer subtypes. There was a large variation in the estimated incidence of RT-related hypothyroidism, with a median estimate of 36% (range 3% to 79%). Reported incidence increased in later publication dates, which was likely related to improved screening and longer follow up. There were a wide variety of predictive metrics used to identify patients at high risk of hypothyroidism, the most common of which were volumetric and mean dosimetrics related to the thyroid gland (Vxx%, Dmean). More recently, there has been increasing evidence to suggest that the thyroid gland volume itself and the volume of the thyroid gland spared from high-dose radiation (VSxx) may better predict thyroid function after RT. There were no identified studies investigating the role of advanced radiotherapeutic techniques such as MRI-guided RT or particle therapy to decrease RT-related hypothyroidism. Conclusions: Hypothyroidism is a common toxicity resulting from therapeutic radiation for head and neck cancer with recent estimates suggesting 40–50% of patients may experience hypothyroidism after treatment. Dosimetric predictive models are increasingly able to accurately identify patients at risk of hypothyroidism, especially those utilizing thyroid VS metrics. Further investigation regarding the potential for advanced radiotherapeutic therapies to decrease RT-induced thyroid dysfunction is needed.

## 1. Introduction

Hypothyroidism is a common complication that can occur after radiation therapy for head and neck cancers. The thyroid gland is particularly vulnerable to the damaging effects of radiation, which can result in a decrease in thyroid hormone production and the development of hypothyroidism. This in turn can lead to impairments in quality of life, including altered general body metabolism, decreased general health perception, worse emotional well-being, cognitive impairment, fatigue, and sexual dysfunction, among others [1,2]. Furthermore, impaired thyroid function has been associated with increased overall risk of mortality [3,4]. Treatment for hypothyroidism often requires lifelong hormone replacement, which itself can contribute to worse perceived quality of life and financial burden not only via direct medication costs but also due to recurrent healthcare visits and laboratory testing. Importantly, thyroid replacement therapy also may be ineffective in as many as 20% of patients due to a variety of causes such as gastrointestinal disease, drug interaction, and non-compliance [5]. Such situations may require more complex treatment strategies to normalize thyroid function. Therefore, the accurate prediction of patients at higher risk of hypothyroidism, and ideally optimization of treatment to decrease such risk, are paramount to improve outcomes for patients receiving radiotherapy for head and neck cancers.

Hypothyroidism has historically been defined using several approaches. Most generically is the clinical definition of hypothyroidism specified by the presence of signs or symptoms related to an underactive thyroid gland such as weight gain, cold intolerance, and fatigue. It also has been described more specifically using biochemical definitions based upon serum levels of thyroxine (T4) and thyrotropin or thyroid-stimulating hormone (TSH) [6]. Classical biochemical hypothyroidism refers to decreased T4 with elevated TSH regardless of clinical symptoms. Biochemically, subclinical hypothyroidism refers to a normal level of T4 in the setting of an elevated TSH [7]. However, specific reported measurements and value ranges may differ according to laboratory measurement techniques, and thus, there may be variation in absolute reported hormone levels across institutions. Given such heterogeneity, for the purposes of this review, we avoided any quantitative analysis of hormone levels and rather categorized hypothyroidism diagnoses based upon the method of evaluation. Any diagnosis that was based purely on clinical symptoms without requirement for biochemical evaluation was considered clinical hypothyroidism and any biochemical presence of underactive thyroid function, including classic subclinical hypothyroidism (elevated TSH with normal T4), was considered biochemical hypothyroidism. Ultimately, regardless of method of evaluation, the decision to prescribe thyroid hormone supplementation is the critical clinical decision-making point that will have implications for patient management strategies and outcomes.

The objective of this review was to systematically analyze the available literature on the incidence, patient- and treatment-related risk factors, and consequences of hypothyroidism after head and neck radiation therapy. The results of this review may have important implications for the management and care of patients undergoing head and neck radiation therapy and may contribute to the development of strategies to minimize the risk of treatment-related hypothyroidism.

## 2. Materials and Methods

### 2.1. Search Protocol and Article Identification

We created a systematic review protocol in accordance with Preferred Reporting Items for Systematic Reviews and Meta-Analysis (PRISMA) guidelines (Figure 1) [8,9]. This protocol has not been registered. We used the Medical Subject Headings (MeSH) and generic search terms to generate the initial PubMed search query on 26 November 2022: (((radiotherapy) OR (radiation therapy)) AND (head and neck)) AND ((thyroid) OR (hypothyroidism) OR (thyroid toxicity) OR (thyroid dysfunction)). Only full research articles were included and no restrictions were made with regard to publication date. Resulting articles were screened initially by abstract review and subsequent review of the full article. Only original articles that were available in English and obtainable for review by the authors were included. Articles were evaluated for inclusion based upon the follow criteria: must report on original research describing the incidence, mechanism, dosimetry, management, or prevention of radiation-relation hypothyroidism for adults receiving RT as a component of treatment for head and neck cancer. Studies reporting on thyroid cancer were excluded. Two reviewers (MKR and KLC) independently reviewed and screened articles for inclusion.

Eighty studies were identified using this initial search and article evaluation. In order to develop a more robust search method, we additionally searched the Google Scholar database using the query “hypothyroidism after radiation therapy” and evaluated the first 500 results. We again applied the evaluation criteria to identify unique articles that were not found in the PubMed search.

Finally, the snowball method was utilized to identity any remaining publications not included in the initial two search protocols [10]. The snowball procedure refers to the evaluation of referenced literature of already identified articles and has been shown to result in the more accurate and exhaustive identification of desired articles [11]. In this review protocol, we reviewed 955 references from the initially included 97 publications and identified 14 additional unique publications.

### 2.2. Article Evaluation and Data Collection

For included articles, we tabulated the following data: first author, publication year, study design (categorized as prospective or retrospective), interventional study (yes/no), study participant (patients or physicians), estimated hypothyroidism incidence (%), sample size, study follow up, head and neck cancer subsite, radiation modality, radiation technique, allowance of multimodality therapy, method for thyroid function evaluation, proposed dosimetric or volumetric risk factor, and other relevant clinical findings.

A study was deemed to be interventional if it described a technique or evaluation of a procedure aimed at preventing or treating RT-induced hypothyroidism. Study follow up was recorded as specifically as possible according to the information presented in the publication. The cancer subsite was categorized as general head and neck cancer, nasopharynx, oropharynx, larynx/hypopharynx, and oral cavity. Radiation modality was recorded as photon, electron, proton, or carbon ion radiotherapy. The technique was categorized as either intensity-modulated radiotherapy (IMRT), MRI-guided radiotherapy, or other. After a full review of all studies, there were none that evaluated MRI-guided radiotherapy, so we report findings as either “IMRT” or “non-IMRT”, which includes 2D and 3D conformal radiotherapy (3DCRT). Multimodality therapy was considered treatment with surgery or systemic therapy in addition to radiation.

Various methods were described to evaluate and characterize thyroid function. We categorized approaches as follows: “biochemical” evaluation refers to the use of laboratory testing to measure hormone levels in the blood, “clinical” refers to the evaluation of symptoms of hypothyroidism or the need for clinical treatment with levothyroxine supplementation. There were also few instances where specific imaging modalities, such as ultrasound or positron emission tomography (PET), were used to evaluate endpoints related to thyroid structure or anatomy such as gland size, radiographic density, or PET avidity. These were characterized as “radiographic” methods and were not direct measures of thyroid function.

There were similarly a wide variety of methods describing predictive models for hypothyroidism. We tabulated all described dosimetric and volumetric parameters described in each article and categorized them in a post hoc fashion. Vxx% refers to the percent volume of the organ receiving a specified dose. Dxx refers to the dose received by a specific volume of the gland. Dmean refers to mean dose. The volume of thyroid gland that was spared by a particular dose (thus receiving that dose or less) was termed VSxx.

## 3. Results

We identified 111 publications that met inclusion criteria according to the search protocol. Characteristics of identified studies are summarized in Table 1. The individual studies are described in Appendix A Appendix A, with study sample sizes provided for each publication.

### 3.1. Estimated Incidence of RT-Induced Hypothyroidism

There was a large range of estimated rates of hypothyroidism following radiotherapy for head and neck cancer reported in the literature, although studies were heterogenous with different patient populations, treatment techniques, sample sizes, and methods/length of follow up. The median reported estimate for all studies was 36% (range 3% to 79%). The median reported mean or median follow-up time was described in 87 papers and was 32 months (IQR 21–50). There appeared to be a relationship between estimated incidence and publication date with more recent publications reporting higher incidence of hypothyroidism (Figure 2). Studies that reported hypothyroidism only based on clinical symptoms (*n* = 2) reported extremely low incidence (mean 4.9%). Additionally, studies with longer reported mean or median follow up were associated with increased estimated incidence of hypothyroidism (Figure 3).

### 3.2. Radiation Modality, Technique, and Dosimetric Predictors of Hypothyroidism

Of the studies that provided specific details regarding radiation modality, the vast majority (98%) reported photon radiotherapy, with a minority including combination (2%) photon/electron therapy. There were no studies identified that reported on the use of particle therapy. There was also a relatively even distribution of radiation techniques, including 2D or 3DCRT (34%) and IMRT (32%), or both (9%) with 24% of studies not specifying technique. It is difficult to ascertain the precise impact of radiation technique on the risk of hypothyroidism in these studies in aggregate due to significant confounding. IMRT was more likely to be utilized in modern studies, and such studies were also more likely to report a higher incidence of hypothyroidism.

Ten studies did include patients that received either 3DCRT or IMRT, of which seven provided a direct comparison of techniques. In six (86%), there were no significant differences in hypothyroidism incidence when comparing individuals receiving IMRT or 3DCRT. Notably, the study by Peng et al. randomized patients between modalities; unfortunately, thyroid function was only assessed clinically, and thus, thyroid dysfunction was exceptionally rare compared to other investigations using robust methods of thyroid evaluation (1.3% vs. 2.9% for IMRT vs. 3DCRT, respectively; *p* > 0.05) [12]. The only other study to report hypothyroidism incidence for patients randomized between modalities was by Murthy et al., who reported an increased risk of hypothyroidism with IMRT (51.1% vs. 27.3%, *p* = 0.021). However, the dose fractionation was different between arms with those receiving 3DCRT being prescribed 70 Gy in 2 Gy fractions and those receiving IMRT being prescribed 66 Gy in 2.2 Gy fractions [13].

Many investigators identified dosimetric parameters or developed dosimetric-based predictive models for the development of hypothyroidism. A summary of various proposed dosimetric planning constraints is summarized in Table 2.

### 3.3. Interventions Aimed at Preventing RT-Induced Hypothyroidism

Four studies were identified that investigated methods to prevent RT-induced hypothyroidism, three of which described a novel free thyroid transfer procedure to transplant the thyroid outside of the radiotherapy field. The proof-of-principle publication showed that it was possible to transfer the gland from the neck to the forearm for patients planned to undergo neck dissection, with successful transfer in 9 of 10 recruited patients, with results maintained at one year for all patients. The procedure added an additional 41.6 min of operative time without any reported additional surgical complications [50,51]. A subsequent publication replicated these results with six patients having the thyroid transferred to the thigh and five patients maintained euthyroid status at median follow up of 8 months [52]. We additionally identified one published prospective protocol investigating the use of amifostine, a recognized radioprotector, during radiotherapy to decrease the risk of developing hypothyroidism [53]. The results of the trial were not able to be identified in a subsequent publication.

### 3.4. Impact of Patient and Physician Factors on RT-Induced Hypothyroidism

Several patient-, treatment- and provider-related factors were associated with reported incidence and management of RT-induced hypothyroidism in the identified body of literature. Some studies (8%) reported that hypothyroidism was more common in females, with several publications suggesting that a decreased average thyroid volume in women was the likely causal factor explaining differences between sexes [31,54]. Furthermore, thyroid gland volume was associated with thyroid function and inversely related to the risk of subsequent hypothyroidism risk following RT. Several studies showed that greater decreases in thyroid volume or density changes following radiation therapy were predictive of subsequent hypothyroidism [30,55,56]. There was mixed evidence regarding the impact of patient age as a risk factor for subsequent hypothyroidism with some studies suggesting that younger age increased risk [57,58] while others suggesting the opposite relationship [27,59]. Combined modality therapy with radiotherapy, surgery, and chemotherapy was frequently reported to be associated with an increased development of hypothyroidism (15%).

There was a small subset of studies that investigated practice patterns of providers with regard to the diagnosis and management of RT-induced hypothyroidism. Lo Galbo et al. conducted a 2006 survey study of physicians in the Netherlands to assess when and how thyroid function tests were conducted after radiotherapy for head and neck cancer [60]. They found that 75% of physicians only tested patients when symptoms were mentioned. When screening, all physicians gathered thyroid-stimulating hormone (TSH), while only 69% collected free T4 as well. The majority of respondents (65%) felt they would appreciate further guidance regarding screening practice. Rønjom et al. studied the impact of variation in target delineation of the thyroid gland among providers and the impact this may have on estimates of hypothyroidism using a normal tissue complication probability (NTCP) model [22]. They found that intra- and inter-observer variability in gland delineation did not have a significant impact on the performance of the toxicity prediction model; however, they do note that for any individual patient, there may be large differences in the estimated risk, and thus, the accurate delineation of the thyroid gland is critical for robust risk prediction.

## 4. Discussion

In this systematic review of the literature, we synthesized and characterized the current evidence regarding the incidence, clinical and dosimetric risk factors, and management and prevention strategies for RT-induced hypothyroidism for adults undergoing treatment of head and neck cancers. We found that the reported incidence of RT-related hypothyroidism has increased over time, which is likely owing to improved screening practice and longer follow up. Although there was wide variation in reported incidence, recent high- quality evidence suggests a rate of 40–50% for all patients with head and neck cancer receiving radiotherapy (Figure 1). Some investigations suggest that RT-induced hypothyroidism occurs relatively shortly after RT (e.g., within months), but others suggest that changes in function may occur for multiple years following treatment [43,44]. Many studies identified clinical and treatment-related factors that predict an increased risk of hypothyroidism, including female sex, decreased thyroid volume, and receipt of multimodality therapy. Furthermore, there is a growing body of research investigating the dosimetric prediction of thyroid dysfunction; early studies largely utilized volumetric thyroid gland parameters, such as V30–50, but more recent investigations suggest that thyroid volume-spared dosimetrics, for example TVS 50–60 Gy, may be the most robust predictors of subsequent toxicity. There were no studies identified by the search protocol that investigated the potential for advanced radiotherapeutic techniques including particle therapy or MRI-guided radiotherapy to decrease the risk of hypothyroidism. These findings may provide guidance for clinicians and researching aiming to improve outcomes for patients receiving RT for head and neck cancers.

The incidence of head and neck cancers has continued to increase with estimates of nearly 660,000 new cases per year, ranking as the seventh most common cancer globally [61]. Of those, it has been suggested that approximately 75% may benefit from radiotherapy [62,63,64]. Therefore, the effective prevention and management of treatment-related toxicity is critical, especially for common side effects which may impact large populations of patients. Hypothyroidism, as evidenced by the robust data in this review, is unfortunately an extremely common result of therapeutic radiation directed to the head and neck. Interestingly, our analysis does suggest that estimates of RT-related hypothyroidism have increased over time with the most recently published papers from 2020 and later estimating 40–50% for all comers, which is likely a result of increased screening (which may detect subclinical hypothyroidism) and longer follow up after treatment. These trends highlight the critical importance of maintaining awareness of this potential side effect to optimally screen patients for potential intervention. To our knowledge, there currently are no consensus guidelines from leading radiotherapy groups regarding optimal screening and management practices for RT-induced hypothyroidism. The data identified in this systematic review suggest a need for such guidelines to standardize practice among providers and improve functional outcomes for patients at large.

Although standardized guidelines regarding screening and management remain elusive, there are robust and growing data regarding the relationship between radiation dosimetry and subsequent thyroid dysfunction. In this review, we identified 38 manuscripts published since 2012 which propose various dosimetric predictors of hypothyroidism. The initial studies in this area largely focused on volumetric and mean constraints related to thyroid gland exposure, such as Dmean, Vxx (%), and Dxx. However, there has been a shift in the most recent analyses, with some reports suggesting that the volume of thyroid spared (VS) metrics outperforms other dosimetrics in terms of toxicity prediction [31,49]. Similar results have also been found in other disease sites such as Hodgkin’s lymphoma where the thyroid gland routinely receives therapeutic-range doses of radiotherapy [65]. Such trends could be explained by the physiologic function of the thyroid gland. As a parallel-functioning structure, the volume of normal glands that can avoid high-dose radiotherapy would intuitively predict the preservation of function. This hypothesis would also be supported by the frequent observation in the included studies that increased thyroid volume itself was protective against subsequent hypothyroidism. As such, we encourage the further research investigation of VS metrics as potential tools to optimize radiotherapy plans and suggest that clinicians consider the implementation of VS metrics as routine components of plan design and evaluation.

Given the clear relationship between radiation dose and risk of thyroid toxicity, improvements in the delivery of conformal radiotherapy will lead to improvements in the therapeutic ratio of head and neck radiation. Over the last several decades, there have been transformative improvements in the available radiation techniques to improve the quality of radiation plans. The most recent widely adopted technology has been IMRT, which allows for inverse-planned beam shaping to prioritize target coverage and minimize dose to organs-at-risk. Many large-scale randomized trials have indeed shown the benefits of IMRT in reducing acute and long-term side effects of radiation across multiple disease sites without compromising oncologic outcomes, including cervical cancer [66], endometrial cancer [67], and prostate cancer [68], among others.

In head and neck cancer, there have also been randomized trials investigating 3DCRT vs. IMRT that have shown a decreased incidence of side effects [69]. One randomized trial reported improvements in many RT-related toxicities, including xerostomia, hearing loss, temporal lobe neuropathy, cranial nerve palsy, trismus, and neck fibrosis with no significant difference in hypothyroidism between groups. However, the absolute incidence of hypothyroidism was 1.3% and 2.9% for 3DCRT and IMRT, respectively, which are both rates that were extremely low compared to other reports. This is likely owing to the fact that investigators only assessed for hypothyroidism with overt clinical symptoms as opposed to biochemical evaluation [12]. In a pooled analysis of two randomized trials, Murthy et al. interestingly found that rates of hypothyroidism were actually higher in patients receiving IMRT, although the authors acknowledge that the results are difficult to interpret for several reasons [13]. Dose and fractionation were different between the IMRT and 3DCRT cohorts; patients in the IMRT group received 66 Gy in 30 fractions and patients receiving 3DCRT were prescribed 70 Gy in 35 fractions. Radiobiologically, the thyroid gland is typically considered a late-responding tissue (therefore, alpha/beta is assumed to be ~3), and the biologically effective doses (BEDs) for these regimens are similar at 114.4 Gy and 116.67 Gy, respectively [70]. Furthermore, and most importantly, IMRT plans on these trials may not have used thyroid avoidance as a treatment planning goal and thus would not be expected to have improved toxicity with regard to the thyroid. Overall, in this review, we were unable to find any high-quality randomized data specifically comparing thyroid toxicity for patients receiving thyroid-sparing IMRT and 3DCRT. Nonetheless, many studies did show improved dosimetry with IMRT which, given the strong evidence discussed previously, would be expected to improve thyroid toxicity profiles through minimization of thyroid dose.

Although IMRT represents a significant advancement in radiation therapy technology, there have been other important developments which may further improve the ability to deliver therapeutic doses to targets while further decreasing the dose to nearby organs. Particle therapy, including proton and carbon ion therapy, is increasingly being investigated as a possible tool to decrease the toxicity of treatment through decreased dose to organs-at-risk. Many prospective and retrospective studies have shown improvements in other disease sites including pediatric cancers [71], esophageal cancer [72], and craniospinal irradiation for solid tumor leptomeningeal disease [73]. A recently completed randomized trial comparing IMRT and intensity-modulated proton therapy (IMPT) for the treatment of advanced oropharyngeal cancers (NCT01893307, PI Frank) will clarify the potential benefits of proton therapy for improving the toxicity of head and neck radiotherapy, including impact on hypothyroidism risk [74].

We also identified several exciting studies that aimed to develop surgical and medical techniques to prevent hypothyroidism after radiation. The free thyroid transfer technique was described by Harris et al. in 2016, in which patients who were planned to undergo neck dissection had their thyroid gland surgically removed and transferred to the forearm. The initial experience with this technique has been promising with the vast majority of patients maintaining a viable transplanted thyroid gland at one year post-procedure [50,51]. A second group reported a similar experience with free thyroid transfer to the thigh, with encouraging initial results [52]. From a clinical perspective, it is important to consider that these early reports to not provide evidence that transplanted thyroid glands maintain long-term endocrine function, and therefore, it is possible that some patients may require thyroxine supplementation despite undergoing the procedure. Additionally, the potential side effects of any additional surgical procedure must be weighed against medical management strategies for hypothyroidism, which in general have a benign side effect profile. As such, while these early studies do provide exciting preliminary evidence that this procedure is possible, longer-term studies with more rigorous evaluation of transplanted thyroid function will be necessary before the routine consideration of this procedure is warranted.

Our search protocol also identified a published clinical trial protocol investigating the use of amifostine to prevent hypothyroidism based upon preclinical data showing its role as a radioprotector [53]. Unfortunately, a subsequent publication of the trial findings was unable to be found. Nonetheless, the proposal to utilize radiomodulating agents, which have been commonly investigated in other settings, may representing a potential way to further decrease the risk of thyroid impairment.

Although this study draws strength from a rigorous search methodology that followed PRISMA guidelines and utilized the snowball technique to increase the search efficacy, it does have limitations. First, only two academic search engines were used; it is therefore possible that some scientific findings were missed that were published in other scientific fora. Second, we limited our search to English articles only and thus may have missed relevant publications in other languages. Third, there was significant heterogeneity in the reporting of some endpoints, which makes it difficult to synthesize findings in a meaningful way. For example, we would have aimed to holistically examine the trends in the detection of subclinical hypothyroidism over time, particularly in the modern era of increased screening. We unfortunately were unable to perform this analysis due to limitations in data reporting. Additionally, it would be clinically important to evaluate the utility of anti-thyroid peroxidase (TPO) antibody levels as a measure or predictor of RT-induced hypothyroidism. However, data regarding anti-TPO levels were rarely reported in the identified body of literature, and we were thus unable to meaningfully synthesize data to address this topic. Last, the search protocol was not replicated multiple times, and it is therefore possible that some individual articles were missed that would have otherwise met inclusion criteria.

## 5. Conclusions

In this systematic review of the literature, we found that RT-induced hypothyroidism is extremely common, with recent estimates of 40–50% utilizing high-quality thyroid function evaluation and long-term follow up. We highlighted several clinical and treatment-related factors that are associated with an increased risk of developing hypothyroidism, including female sex and decreased normal thyroid volume. Furthermore, we summarized the evidence regarding dosimetric-based toxicity prediction and recommend that clinicians consider the use of thyroid volume spared dosimetrics to optimize radiation plans, as these metrics appear to be the most robust predictors of RT-induced hypothyroidism in modern studies. Ongoing research will clarify the potential role for advanced techniques such as particle therapy and free-thyroid transfer to further reduce the risk of hypothyroidism. These findings can be used by providers and researchers alike to improve the care of patients with head and neck cancer by minimizing the symptom and financial burden associated with lifelong hypothyroidism after treatment.

## Figures and Tables

**Figure 1 cancers-15-04321-f001:**
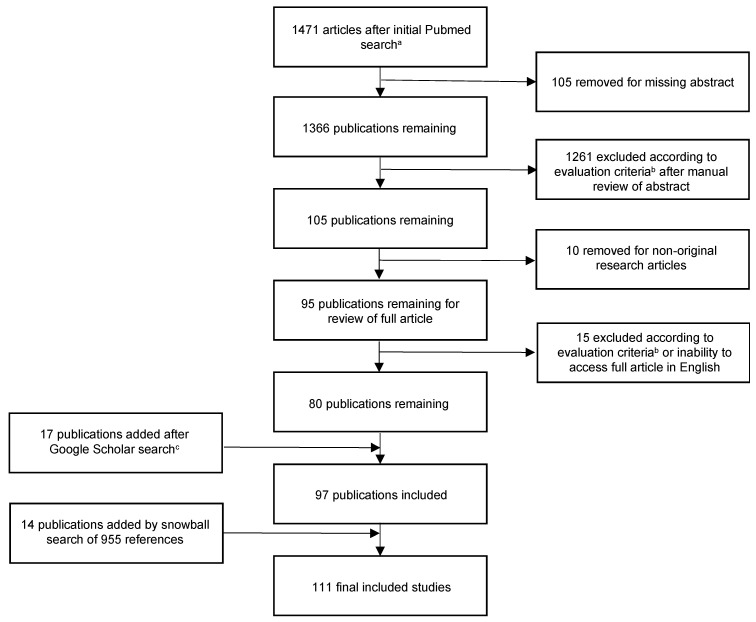
Flow diagram depicting the article identification protocol. ^a^ Initial search criteria in PubMed, as of 26 November 2022: (((radiotherapy) OR (radiation therapy)) AND (head and neck)) AND ((thyroid) OR (hypothyroidism) OR (thyroid toxicity) OR (thyroid dysfunction)); ^b^ Article evaluation criteria: must be original research article describing incidence, mechanism, dosimetry, treatment, or prevention of radiation-related hypothyroidism for adults receiving RT for the treatment of head and neck cancers. ^c^ Evaluated and included non-duplicate entries from the 500 hits from the Google Scholar query “hypothyroidism after radiation therapy”.

**Figure 2 cancers-15-04321-f002:**
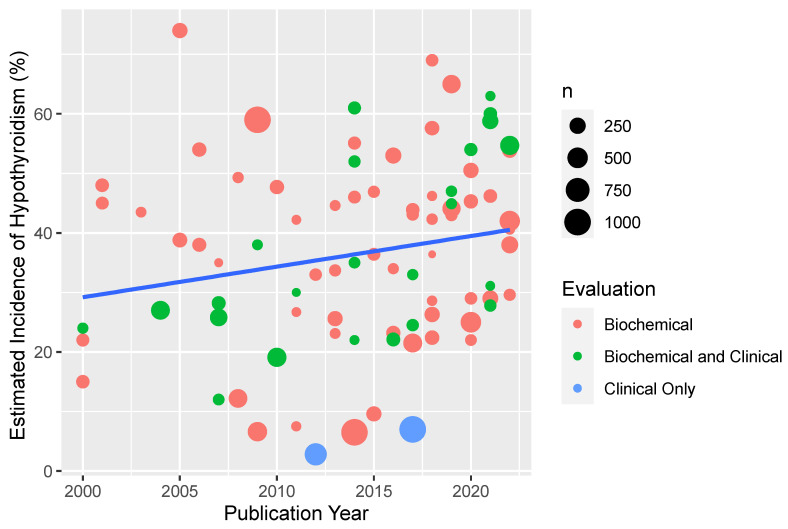
Estimated incidence of RT-related hypothyroidism for included studies displayed over time by year of publication. Each dot represents a study, with size weighted according to the sample size of patients reported in the publication. The color of the dot reflects the method of thyroid function evaluation. Note: studies published before 2010 were excluded from this figure to improve clarity. Studies using radiographic evaluation of the thyroid gland also included biochemical evaluation and were considered as “biochemical” in the figure.

**Figure 3 cancers-15-04321-f003:**
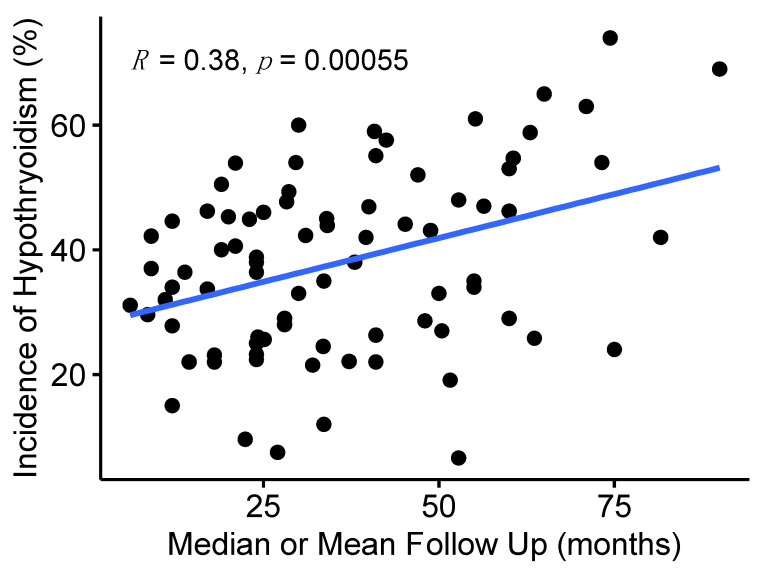
Correlation between estimated incidence of hypothyroidism and reported mean or median follow up of selected studies. Studies using only clinical evaluation of thyroid function (*n* = 2) and one outlying retrospective study published in 1984 with small sample size, long follow up, and low incidence of hypothyroidism were excluded to improve clarity and accuracy of the figure.

**Table 1 cancers-15-04321-t001:** Characteristics of included studies, with summary statistics of reported hypothyroidism incidence displayed by subgroup.

Factor	Count (%) N = 111	Estimated Hypothyroidism Incidence (%), Reported as Median (IQR)
Publication Year		
2000 and prior	19 (17)	34 (21–43)
2001–2005	8 (7)	44 (40–47)
2006–2010	15 (14)	31.6 (20.8–45.3)
2011–2015	22 (20)	33.3 (22.8–45)
2016–2020	32 (29)	42.3 (25–46.2)
2021–2022	15 (14)	42 (30.4–54.3)
Design		
Prospective	31 (28)	37 (27.4–47)
Retrospective	80 (72)	36.1 (24–46)
Interventional		
Yes	4 (4)	-
No	97 (96)	-
Mean/Median Follow Up (months)		
Min	6	-
First Quartile	21	-
Median	32	-
Third Quartile	50.2	-
Max	120	-
Unspecified/NA	24	-
Sample size	Median 102 (62–184.75)	-
Participant Type		
Patients	99 (98)	-
Physicians	2 (2)	-
Cancer Site		
General Head and Neck	64 (58)	36.7 (26–44.7)
Nasopharynx	28 (25)	35 (23–46.2)
Oropharynx	6 (5)	57.5 (35.3–62.5)
Larynx/Hypopharynx	10 (9)	39.5 (28.7–48.7)
Oral Cavity	3 (3)	17.9 (16.1–19.7)
Radiation Therapy Modality		
Photons	87 (78)	-
Photons/Electrons	2 (2)	-
Particle Therapy	0	-
Unspecified	22 (20)	-
Radiation Therapy Technique		
IMRT	36 (32)	42 (27.4–49.1)
Non-IMRT (2D or 3DCRT)	38 (34)	35 (21.8–41.6)
IMRT and 3DCRT	10 (9)	46.5 (27.5–50.8)
MRI-guided radiotherapy	0	-
Unspecified/NA	27 (24)	35 (23.9–48.2)
Multimodality Therapy		
All therapies allowed	89 (80)	36.4 (24.8–45.7)
Surgery excluded	13 (12)	38.8 (30–47)
Surgery required	4 (4)	38.6 (26.2–50.5)
Unspecified/NA	5 (5)	36.4 (25.4–48.2)
Thyroid Evaluation		
Biochemical	73 (66)	38.8 (26.7–46.2)
Clinical	2 (2)	4.9
Biochemical and Clinical	32 (29)	31.1 (23.9–49.5)
Biochemical and Radiographic	2 (2)	42
Unspecified/NA *	2 (2)	-

Abbreviations: 2D and 3DCRT; two-dimensional and three-dimensional conformal radiation therapy, IMRT; intensity-modulated radiation therapy * Refers to studies investigating physician practices regarding screening for hypothyroidism and thus were considered Unspecified/NA.

**Table 2 cancers-15-04321-t002:** Proposed dosimetric and organ-at-risk parameters to predict risk of RT-induced hypothyroidism.

Author	Year	Proposed Predictive Dosimetric and OAR Parameters
Boomsma, M.J. [14]	2012	Dmean, thyroid volume
Rønjom, M.F. [15]	2013	25% risk of hypothyroidism for D10cc = 26 Gy, D15cc = 38 Gy, D20cc = 48 Gy, D25cc = 61 Gy
Bakhshandeh, M. [16]	2013	D50 < 44 Gy
Huang, S. [17]	2013	Dmean, V40
Murthy, V. [13]	2014	D100 was significant predictor on MVA
Akgun, Z. [18]	2014	Dmean, thyroid volume, V30
Kim, M.Y. [19]	2014	V45 Gy < 50%
Chyan, A. [20]	2014	Dmean < 49 Gy, VS 45 Gy > 3cc, VS 50 Gy > 5cc, V50 Gy < 45%
Fujiwara, M. [21]	2015	Dmean < 30 Gy
Rønjom, M.F. [22]	2015	Dmean, thyroid volume
Pil, J. [23]	2016	Dmean, thyroid volume
Luo, R. [24]	2016	V50 Gy
Lee, V. [25]	2016	VS 60 Gy > 10cc, VS 45 Gy > 5 cm
Ling, S. [26]	2017	D50 < 50 Gy, V50 < 50%, Dmean <54.58 Gy
Sommat, K. [27]	2017	V40 < 85%
Zhai, R. [28]	2017	V45 < 50%; V50 < 35%
Sachdev, S. [29]	2017	V50 Gy < 60%
Lin, Z. [30]	2018	Dmean, D50
Xu, Y. [31]	2018	V50 < 54.5%, Dmean < 51.6 Gy
Lertbutsayanukul, C. [32]	2018	VS 60 Gy > 10cc
El-Shebiney, M. [33]	2018	V30 < 42.1%
McDowell [34]	2018	V50 > 90%. V45 > 99%
Luo, R. [35]	2018	V50 Gy, Dmax pituitary
Kamal, M. [36]	2019	Dmean, thyroid volume
Prpic, M. [37]	2019	Dmin, thyroid volume
Huang, C. [38]	2019	V25 < 60%, V35 < 55%, V45 < 45%
Lin, A.J. [39]	2019	V50 Gy < 75%
Peng, L. [40]	2020	V30, V60, thyroid volume
Kinclová, I. [41]	2020	V40 < 85%
Zhou, L. [42]	2020	V50 Gy, thyroid volume < 12.82cc
Zhu, M.Y. [43]	2021	V35
Inoue, E. [44]	2021	V45 Gy > 67%
Lu, H.H. [45]	2022	Dmean, thyroid volume
Zhai, R. [46]	2022	V40 < 80%
Pal, S.K. [47]	2022	D50, DmeanV50 Gy
Jia-Mahasap, B. [48]	2022	VS 50 Gy
Chow, J.C.H. [49]	2022	VS 60 Gy

Abbreviations: Vxx% refers to the percent volume of the organ receiving a specified dose. Dxx refers to the dose received by a specific volume of the gland. Dmean refers to mean dose. VSxx represents the volume of thyroid that was spared by a particular dose (thus receiving that dose or less).

## Data Availability

Data will be made available upon reasonable request to the corresponding author.

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
