# Peer review of "Hypothyroidism following Radiotherapy for Head and Neck Cancer: A Systematic Review of the Literature and Opportunities to Improve the Therapeutic Ratio"

_cancers, 2023, doi:10.3390/cancers15174321_

Round 1
Reviewer 1 Report
Hypothyroidism following radiotherapy for head and neck cancer: a systematic review of the literature and opportunities to improve the therapeutic ratio, review article proposed by Rooney et al. addresses a topical and certainly neglected topic in dealing with the toxicities associated with head and neck cancers: radiation-induced hypothyroidism. The review is detailed, well organized and interesting, including the patient selection algorithm, two tables and two figures. I appreciate the work of synthesizing the dosimetric data regarding hypothyroidism. Mentioning two different radiotherapy regimens, one with fractionation of 2 Gy and another with doses per fraction of 2.2 Gy using the IMRT technique, I would also include a short radiobiological comparison (if possible mentioning the alpha/beta ratio reported for hypothyroidism). The interest in spared volume of thyroid (another interesting topic developed by the author) also justifies the mention of this trend in the conclusions.
Reviewer 2 Report
The authors present an interesting systematic review on the risk of hypothyroidism after radiotherapy for head and neck cancers. The paper is very interesting, but there some points that could be improved.
145 imaging modalities, such as ultrasound or positron emission tomography (PET), were used to evaluate thyroid function. These were characterized as “radiographic” methods. Comment: None of these imaging techniques are able to assess the thyroid function, ie the secretion of thyroid hormones, therefore the radiographic category should not be mentioned as a separate category.
152 The volume of thyroid gland that was spared by a particular dose (thus receiving that dose or less) – spared means it did not receive the full dose.
Results
The overall number of the patients included in the studies should be provided, as well as the number in the distinct cancer categories (it may be included in Table 1)
In table 1, in Mean/Median Follow Up (months) – what does the number 24 mean in Unspecified/NA? no. of studies with unspecified follow-up?
In table 1, what does Biochemical and clinical evaluation mean? That some patients were evaluated with thyroid hormone measurement and others were just treated for symptoms, but not biochemically assessed? What does Biochemical and radiographic mean? That some patients were deemed to be hypothyroid just on imaging characteristics (which is incorrect?). If in this category all the patients were biochemically tested, I suggest to put those studies in the Biochemical category (in the table 1 and figure 2)
Figure 1 Legend: b. is missing in the text of the legend.
Row 170 The median reported estimate for all studies was 36% (range 3% 170 to 79%). -in the abstract it is written 2 – 79%, please correct.
Table 2 is not clear. Please explain all the abbreviations in a legend and try to be more clear in the description of the results. For example, it is not clear what means V35, In the study of Zhu or V45Gy<50% or D10cc? In the studies in which only the parameters were described, there are no cuttoffs calculated for these parameters and the risk of hypothyroidism?
235 Many studies reported that hypothyroidism was more common in females: please provide numbers from the studies (percentages )
237 Unclear phrase: there were several studies suggesting that larger thyroid volume (how large?, cuttoff?) decreases (the risk?) or density changes (in which way? Decreases or increases? examples) following radiation therapy were predictive of subsequent hypothyroidism
242 Combined modality therapy with radiotherapy, surgery, and chemotherapy was frequently reported to be associated with increased development of hypothyroidism – how frequently? Percentages ?
Describe NTCP abbreviation
256 however, they do note that for any given patient they may be large differences (I would suggest: for an individual patient there may be large differences) in estimated risk and thus accurate delineation of the thyroid gland is critical for treatment optimization and robust model performance (risk prediction?).
265 recent high- quality evidence suggests a rate of 40-50% for all patients with head and neck cancer receiving radiotherapy (citation needed, and the median follow-up duration in those studies ). When do most of the cases occur? (Some studies – eg Pal et al, 2022 - show that most of the cases occur during the first year after radiotherapy – this is an interesting information that should be mentioned, if it was sustained also in other studies)
Discussion: explain in few words for non-radiotherapy specialists what mean the parameters you mention (eg, V30-50, TVS 50-60Gy)
285: Interestingly, our analysis does suggest that estimates of RT-related hypothyroidism have increased over time with most recently published papers from 2020 and later estimating 40-50% for all comers, likely a result of increased screening (which identifies also subclinical hypothyroidism) and longer follow up after radiation treatment.
It would be interesting to mention the percent of patients with subclinical hypothyroidism in the more recent studies with biochemical screening, since most of them do not need thyroxine treatment (except women seeking to obtain a pregnancy, some young patients < 30 years, those with TSH >/= 20 uIU/mL, or those with significant symptoms). However they need monitoring for the potential evolution to overt hypothyroidism. See Thyroid hormones treatment for subclinical hypothyroidism: a clinical practice guideline, BMJ 2019
315 Over the last several decades, there have (been) transformative improvements in the available radiation techniques to improve the quality...
Minor English corrections are needed - mostly spelling errors
Reviewer 3 Report
The paper is focused on the problem of hypothyroidism induced by radiotherapy for head-and-neck cancers. It is interesting because we usually concentrate on the increased risk of secondary thyroid cancer in such patients. In fact we can safely assume that hypothyroidism is much more common complication, despite some uncertainty regarding their incidence. I presume that all the authors are experts in radiotherapy and oncology as some parts of the manuscript suggest lack of expertise in endocrinology. An incorporation of an endocrinologist would be beneficial. Especially some parts of the introduction should be rewritten, because quite unnecessary authors tried to convince potential readers that hypothyroidism is a grave complication. For example, they wrote (lines 60-62): ”Treatment for hypothyroidism often requires lifelong hormone replacement… agreed …, which itself can contribute to worse perceived quality of life and potential financial toxicity”. Financial toxicity sounds strange to non-native ears but a quick google search gives the following information: “How much does levothyroxine cost? Generic levothyroxine, in most cases, will be covered by both private health insurance policies and Medicare/Medicaid, but for those who do not have a health insurance plan, then it is best to budget at least $3 to $15 per 30 tablets. This cost will greatly depend on the pharmacy you use.” (www.howmuchisit.org/how-much-does-levothyroxine-cost/). Of course this information is specific to the U.S. but in developed countries, those with radiotherapy wildly available, levothyroxine monthly cost for the patient is usually even lower than $10. It is generally affordable and the actual clinical problem with levothyroxine therapy is that it is not always effective. In about 10% of hypothyroid patients some clinical symptoms persist despite normalization of fT4 levels. Some patients need combination of T4 and T3 and we don’t know how to identify these patients in advance. There are also some interactions with other medications that can prevent levothyroxine absorption (e.g. IPP) or affect these medications (e.g. warfarin).
Interestingly, it seems that the analyzed papers, partly due to the keywords used, were also written from a radiotherapist point of view. For a thyroidologist it seems quite obvious that anti-thyroperoxidase antibodies should be included in methods predicting developing hypothyroidism after head and neck radiotherapy. Simply because a patient with high anti-TPO levels is probably on their way to develop clinically overt chronic thyroiditis and their thyroid already have a limited ability to synthesize T3 and T4. This is especially important in countries with high-normal iodine supply.
At lines 73-75 the authors wrote “Biochemically, subclinical hypothyroidism refers to a normal level of T4 in the setting of an elevated TSH.[6] However, these definitions themselves can be varied as specific laboratory measures may have different normal ranges for each hormone.” The former statement is true but the latter is false. Of course laboratories use various kits for the measurements, and consequently they use different normal ranges but that does not affect the definition of subclinical hypothyroidism which based not on any specific concentration of TSH or fT4. So, we regard TSH concentration as elevated only if it exceeds upper limit of the normal range specific to the method used. Again, the actual clinical problem is whether and when to start the treatment with levothyroxine in patients with subclinical hypothyroidism but not how to identify this condition.
Lines 145-147: “There were also instances where specific imaging modalities, such as ultrasound or positron emission tomography (PET), were used to evaluate thyroid function. These were characterized as “radiographic” methods” – well, this is an endocrine heresy. It is not possible to evaluate thyroid function with imaging studies alone. Actually we fail medical students for such statements. I do agree that functional imaging, like 131I scintigraphy or PET, correlate with thyroid function but they can never should be used for diagnosing abnormal thyroid function. And the idea of using ultrasound imaging after RT to assess functional status of the thyroid is totally absurd. In my opinion such studies should be excluded from the analysis as unreliable.
The authors reported papers describing a novel free thyroid transfer procedure to transplant the thyroid outside of the radiotherapy field. I found it interesting but there are two issues. First, the papers are reported twice: in section 3.3 and then again in a paragraph in the discussion (lines 355-367) with most information just repeated. Second, I miss some critical discussion of these reports. The concept resembles a similar procedure used for parathyroid glands but parathyroids are smaller and their hypofunction is much more difficult to compensate. Personally, I’d preferred long-life therapy with levothyroxine over having my forearm cut. More importantly, these reports only showed that transplantation of the thyroid lobe had not underwent atrophy after several months but did not prove that such a transplant was effective as alone source of thyroid hormones (the other lobe was left intact). So I would appreciate more critical analysis in the discussion instead of repetition.
Minor remark: Reference no. 13 – the doi indicates some other paper
Despite all the criticism expressed above – which a duty of the reviewer – I found the paper interesting and important. It well signals that there should be better understanding of hypothyroidism as a possible complication of head-and-neck radiotherapy, especially as the early identification of hypothyroidism solely based on clinical signs and symptoms may be challenging even for an endocrinologist. I up vote the idea of guidelines here.
I am not a native speaker so my comments on the text written by native American English speakers may be perceived as ridicule. Anyway, I am brave enough to suggest that the word 'receipt' in the sentence "Multimodality therapy was considered receipt of surgery or systemic therapy in addition to radiation" (lines 139-140) sounds strange and could be replaced for better understanding by non-native readers
Round 2
Reviewer 2 Report
The revised version of the paper is ready to be published